# OpenReview forum: "Honesty to Subterfuge: In-Context Reinforcement Learning Can Make Honest Models Reward Hack"
_NeurIPS.cc/2024/Workshop/SafeGenAi — SafeGenAi Poster_

### Official Review · Reviewer_FTPq · 2024-10-09
**Inspiring findings, nice workshop paper!**

**Rating:** 8
**Confidence:** 3

**Review:**

### Summary
This paper explores the application of In-Context Reinforcement Learning in offline reinforcement learning settings, specifically focusing on the model's ability to discover specification gaming strategies through iterative reflection. By comparing ICRL with traditional Single-Episode Generation and expert iteration techniques across a series of increasingly challenging tasks, the study illustrates how ICRL enhances model performance and propensity for discovering rule-exploiting behaviors without explicit training on these tactics. I believe this paper makes an interesting point while we heavily rely on LLMs. As a workshop venue, I am confident that this topic would lead to thoughtful discussions among peers.

---

### Official Review · Reviewer_2LPc · 2024-10-09
**shows in-context iterative reflection can outperform standard expert iteration in some limited settings**

**Rating:** 6
**Confidence:** 3

**Review:**

The authors propose in-context iterative reflection / in-context reinforce learning. They empirically demonstrate that compared to standard expert iteration iteration method, their approach may increase gpt-4o-mini's propensity to learn reward-hacking strategies. They showcase that their approach allows learning reward-hacking policies that are rarely explored during zero-shot usage.

Strength: They provided analysis on settings without finetuning (purely LLM reflection) and settings with supervised finetuning. They give interesting insights on the model's chain-of-thought reasoning, specifically after training with their approach, the model's CoT reasoning becomes more misaligned than the baseline model.

Weakness: the experiments are of small scale and only one LLM (gpt-4o-mini) is used in expert iteration experiments. This limits the scope of the work and leaves open the questions of whether similar results can be obtained on other LLMs. Additionally, the authors train using selected samples that feature reward hacking, which should be only a small fraction of training set in more realistic settings.

Additional comments: in Figure 4, the results are for 512 10-episode rollouts and 128 5-episode rollouts. It would strengthen the paper to present error bars from in this figure. It may improve the paper to dive deeper into experiments related to CoT reasoning and highlight these findings in the introduction.

---

### Official Review · Reviewer_Qcpt · 2024-10-09
**The paper studies the effect of incorporating In-Context Reinforcement Learning into expert iteration on a curriculum of gameable tasks for LLMs. But the experiments are not sufficient due to the limited number of tasks and the small scale of LLMs.**

**Rating:** 5
**Confidence:** 2

**Review:**

### Quality, Clarity, Originality and Significance:
The paper is well-written, with a clear motivation for the problem and the method. It provides a detailed explanation of their experiments.

### Strengths
- The paper provides new insights that through pure iterative reflection during inference, current LLMs are able to learn specification-gaming strategies.

### Weaknesses
- The paper draws its conclusion based on five tasks, additional tasks are needed to further validate the findings.
- One of the key findings 'Incorporating ICRL into expert iteration on a curriculum of gameable tasks may increase gpt-4o-mini’s propensity to learn specification-gaming policies' is stated to be valid only under certain conditions, which diminishes the overall significance of the conclusion. Additional experiments on other LLMs are necessary.